# Trust the PRoC3S: Solving Long-Horizon Robotics Problems with LLMs and Constraint Satisfaction

**Aidan Curtis***, **Nishanth Kumar***, **Jing Cao, Tomás Lozano-Pérez, Leslie Pack Kaelbling**

MIT Computer Science and Artificial Intelligence Laboratory
{curtisa, njk, jingcao, tlp, lpk}@csail.mit.edu.

**Abstract:** Recent developments in pretrained large language models (LLMs) applied to robotics have demonstrated their capacity for sequencing a set of discrete skills to achieve open-ended goals in simple robotic tasks. In this paper, we examine the topic of LLM planning for a set of *continuously parameterized* skills whose execution must avoid violations of a set of kinematic, geometric, and physical constraints. We prompt the LLM to output code for a function with open parameters, which, together with environmental constraints, can be viewed as a Continuous Constraint Satisfaction Problem (CCSP). This CCSP can be solved through sampling or optimization to find a skill sequence and continuous parameter settings that achieve the goal while avoiding constraint violations. Additionally, we consider cases where the LLM proposes unsatisfiable CCSPs, such as those that are kinematically infeasible, dynamically unstable, or lead to collisions, and re-prompt the LLM to form a new CCSP accordingly. Experiments across simulated and real-world domains demonstrate that our proposed strategy, PRoC3S, is capable of solving a wide range of complex manipulation tasks with realistic constraints much more efficiently and effectively than existing baselines. Website: https://proc3s.csail.mit.edu.

**Keywords:** LLMs for planning, task and motion planning, constraint satisfaction

## 1 Introduction

Recent progress on large-scale foundation models, particularly large language models (LLMs) and vision-language models (VLMs), has enabled a variety of flexible and general-purpose decision-making systems for robotic tasks [1, 2, 3, 4, 5, 6, 7]. These systems leverage few-shot prompting as well as the commonsense and sequence prediction abilities of LLMs and VLMs to output sequences of robotic skills that achieve a wide variety of goals. Such systems are generally more capable at handling open-world environments than classical systems, like task and motion planners (TAMP) [8], since they often do not require hand-specified symbolic components (e.g. predicates and operators), and can perform tasks specified directly in natural language or images.

While foundation models have been applied to a range of robotic tasks, these tasks share many common simplifying and limiting assumptions. In most cases, the system is provided with a fixed set of discrete skills such as `robot.move_to_door()` or `robot.pick_can()` and asked to perform tasks that simply require composing these skills in a particular order. Discrete skills with minimal to no control over the skill outcome may be sufficient for simple tasks and settings, but are insufficient for domains with complex constraints or goals that depend on continuous properties or relationships, which are common in robotics. For instance, consider the goal shown in the Arrange-YCB domain from Figure 4 in which the robot is tasked with packing a set of YCB objects into a small region. Collision constraints on placing locations restrict the space of possible picking grasps on each object. A single monolithic `robot.pick_can()` skill is not sufficient in this context, regardless of

---

*Equal contribution.

8th Conference on Robot Learning (CoRL 2024), Munich, Germany.

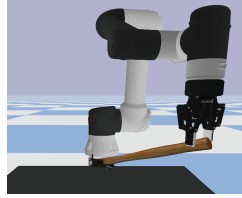 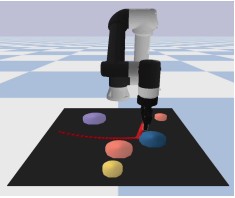 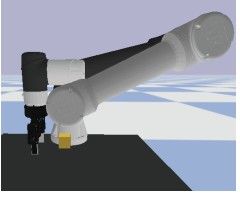 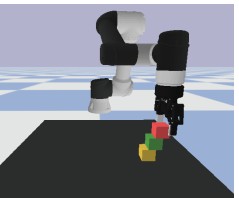

Grasp Failures     Obstacle Collisions     Kinematic Infeasibility     Lack of Stability

Figure 1: Illustration of some common constraints in robotic domains.

that skill's success rate in isolation. Additionally, goals that specify continuous properties or relationships between objects require actions with continuous parameters. For example, a goal such as "place the object in the top right corner" or "draw a star" displayed in Figure 4 require the planning system to have fine-grained access to specific skill outcomes to avoid the kinds of failures displayed in Figure 1. Such access requires *parameterized* skills [9, 10].

We seek to address these limitations and enable planning systems based on foundation model to address complex robotic tasks with realistic constraints using parameterized skills. Towards this goal, we take inspiration from TAMP, and separate planning into two distinct phases [11, 12, 8]. In the first phase, we ask an LLM to generate a program that takes in some continuous parameters as input and produces a sequence of skills with all their parameters specified. We also prompt the LLM to associate a sampling function with each continuous parameter. The result of this phase is thus a kind of continuous constraint satisfaction problem (CCSP) [13, 14, 8, 15]: the solver must now find values for the continuous parameters of each skill such that the proposed sequence of skills can be executed to achieve the goal without violating any constraints in the world (Figure 1). The second phase attempts to solve this CCSP via a simple generate-and-test procedure. If it is unable to find a setting for all the continuous parameters that leads to goal achievement, it reports the failure modes encountered and asks the first stage for a new, different sequence that resolves this issue. This planning process, similar to TAMP, is carried out entirely within a simulated world model, which can be constructed from visual inputs using pretrained perception models [16]. Once a viable plan is found, we execute it in the real environment and replan if necessary.

We evaluate our approach, Planning for Robots via Code for Continuous Constraint Satisfaction (PRoC3S), on a range of challenging robotic tasks in three different simulated domains, and one real-world domain. In particular, we measure the agent's success rate at completing the tasks involving drawing, rearranging, stacking, and packing objects into configurations specified by a natural language goal. In contrast to classical planning systems, our approach demonstrates the ability to satisfy a diverse set of natural language goals without symbolic predicates or operators. Our system also exhibits greater robustness to real-world constraints than existing methods that apply foundation models to robotics problems.

## 2   Related Work

The traditional approach to solving long-horizon robotics problems with complex constraints is task and motion planning (TAMP), which combines both a higher-level logical planner and a set of low level parameterized skills [8, 11, 16, 12, 17]. While this is a powerful framework that enables zero-shot generalization to new problems, the set of capabilities of the system are limited to goals that can be expressed using some set of pre-specified predicates, and via a sequence of pre-specified symbolic operators. Some recent work has used LLMs to guide search and translate natural language goals into logical ones, but still require manual specification of operator preconditions and effects or goal predicate classifiers [18, 19, 20]. By contrast, our approach is able to accomplish tasks where specifying symbolic predicates and operators is either challenging or impossible (e.g. the tasks from our 'Drawing' domain in Section 5).

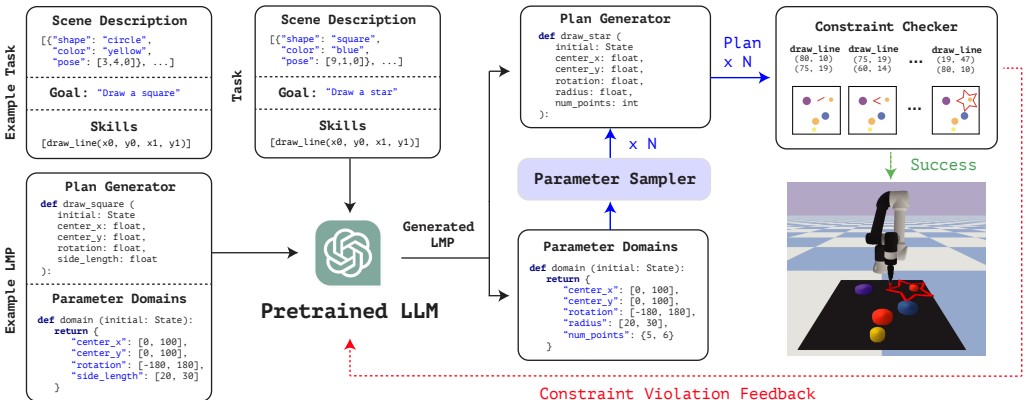

Figure 2: Overview of PRoC3S. An LLM is prompted with an example initial state, goal, LMP, and associated LMP domain for drawing a square. When prompted with a new state and goal for drawing a star, the language model outputs a new LMP and associated domain. We then sample inputs to the function and test them against a set of pre-specified constraints via a simulator. If no satisfying assignment is found after N samples, we feed back the primary failure modes to the LLM to generate an updated LMP and domain.

Recent advances in LLMs have enabled LLM-based planning systems wherein the available skills are described in natural language and their effects on the world need not be explicitly defined [21]. One of the first such approaches sequences discrete object-specific skills to satisfy rearrangement goals [4]. Many follow-up papers have extended this framework to handle longer-horizon tasks with temporal dependencies [7, 22, 2]. Others have made these action-selection strategies more reactive by reprompting with feedback from the environment or running optimization over skill sequences [23, 6, 24, 25]. Although more flexible than TAMP, these approaches can only make use of discrete skills with no continuous input parameters, which greatly restricts the class of problems they can solve. Some recent work has sought to remedy this by having the LLM generate code that transforms environment parameters into action input parameters via a code interpreter [1]. A downside of this method is that it does not properly handle kinematic, collision, or dynamic constraints of the robot's embodiment. Other approaches have the LLM directly output continuous action parameters and use environmental feedback to adjust those parameters to satisfy encountered constraints [3, 24]. These methods directly rely on an LLM or VLM to resolve constraints, which assumes these model are capable of complex geometric and physical reasoning they are generally not trained for.

Some existing work has taken environmental constraints into account when executing skills by building more context-aware skill primitives [5, 26] or used solvers to satisfy LLM-suggested constraints for non-robotic domains [27, 28]. We instead focus on long horizon robotic manipulation problems with continuously parameterized skills and temporally dependent constraints.

## 3 Problem Setting

We consider a robot planning *task* with object-oriented states and parameterized skills defined by the tuple $\langle \ell_G, \mathcal{S}, s_0, \Phi, f, C \rangle$. Here, $\ell_G$ is a natural language goal represented by a string corresponding to some unknown goal condition defined over the state space $G \subseteq \mathcal{S}$.

$\mathcal{S}$ is the robot's state-space. We assume the state is *object-oriented* and fully-observable: i.e., it is factored into a discrete set of objects, each with a set of attributes that may be discrete, continuous, or a string. We assume each object is an instance of a class in the object-oriented programming sense. Given a finite set of objects, the state space $\mathcal{S}$ is defined by the possibly infinite set of assignments to these object's attributes. The initial state $s_0$ is a collection of instantiated objects. For example, an initial state of our Arrange-Blocks environment with two objects might be:

```
{"o1": Object(cat="block", color="yellow", pose=[0.04, -0.36, 0.02, 0.0, -0.0, -0.0])
 "o2": Object(cat="bowl",  color="green",  pose=[-0.14, -0.35, 0.03, 0.0, -0.0, 0.0])}
```

We also assume that the robot has access to a set of lifted parameterized skills $\Phi$. Each lifted skill $\phi_{\Lambda,\Theta} \in \Phi$ (e.g. `Pick([obj], [x, y, z, r, p, y])`) has a name (i.e., `Pick`), a natural language description (e.g. "Move the robot's gripper to location x, y, z and close the gripper."), and a tuple of discrete $\Lambda$ (i.e., `[obj]`) and continuous parameters $\Theta$ (i.e., `[x, y, z, r, p, y]`) that govern the behavior of the skill. Each $\lambda \in \Lambda$ has a discrete domain and each $\theta \in \Theta$ has a continuous domain that are skill-specific. For example, the `obj` parameter would have a domain consisting of the names of all the objects in the current world state (i.e., banana, spam, etc. in the Arrange-YCB domain), while the `[x, y, z]` parameters would have domains corresponding to the edges of the table surface, and the `[r, p, y]` parameters are constrained to be within $[0, 2\pi]$ radians. A lifted skill $\phi$ can be *grounded* $\underline{\phi}$ by selecting values for each of the parameters, resulting in a possibly infinite set of ground skills $\underline{\Phi}$. Ground skills can be executed from any $s \in \mathcal{S}$ and terminate upon reaching a skill-specific termination condition, which will result in a new state $s' \in \mathcal{S}$, where it's possible that $s = s'$. For instance, a `Pick(banana, [0.1, 0.2, 0.16, \pi/2, 0, \pi/4])` attempts a grasp relative to the banana pose. It can be executed from any state in the Arrange-YCB environment, and may or may not pick up the banana depending on these parameters. Lastly, as in any planning system, the robot is given access to a transition model $f : \mathcal{S} \times \underline{\Phi} \to \mathcal{S}$. In our case, this is implemented with a physics simulator.

Lastly, we assume a finite set of user-defined constraints $C$. Each constraint $c_i \in C$ has a corresponding natural language description of what violating the constraint entails (e.g. "Pose is not reachable by gripper"), and a classifier $c_i^\psi : \mathcal{S} \to \{\text{true}, \text{false}\}$ mapping a state $s \in \mathcal{S}$ to a boolean value indicating whether or not the particular constraint is violated. Constraints may be induced by the kinematics of the robot, collisions with the environment, or dynamic properties like stability of the robot or objects the robot is interacting with, and are common across a wide range of robotic tasks. To check whether a constraint has been violated, we will set our simulator $f$ to a particular state and call the constraint's classifier function[2].

The robot's objective is to find a plan $[\underline{\phi}_0, \underline{\phi}_1, ..., \underline{\phi}_K]$ defined by a sequence of ground skills such that: (1) sequential execution of the plan from $s_0$ yields a state sequence $[s_1, s_2, \ldots, s_{K+1}]$ such that $s_{K+1} \in G$, and (2) no state $s \in [s_1, s_2, \ldots, s_{K+1}]$ violates a constraint function (i.e., $\nexists s \in [s_1, s_2, \ldots, s_{K+1}] : \exists c_i \in C : c_i^\psi(s) = \text{true})$ [3].

## 4  Method

Following previous work [1, 3, 2], we solve planning tasks by querying an LLM to directly generate a sequence of skills that achieve the goal from the initial state $s_0$. However, generating a plan with an LLM is a challenging problem because it involves both correctly sequencing skills together, and also finding a specific setting of all the continuous parameters that enables the plan to achieve the goal. For instance, consider the "Draw a star" task depicted in Figure 4. Here, the robot is provided with a `robot.draw_line(`$x_0, y_0, x_1, y_1$`)` skill, that draws a straight line between the points $(x_0, y_0)$ and $(x_1, y_1)$ respectively. To successfully accomplish the task, the robot must invoke this skill at least 5 times in sequence. Moreover, it must specify at least 20 continuous parameters (10 pairs of $(x_0, y_0)$ tuples) such that the shape can be drawn without violating collision or reachability constraints.

To address these challenges, we take inspiration from TAMP in two significant ways: (1) we provide the LLM with access to code for a set of *samplers* [29, 8], $\Sigma$, to help it sample continuous parameters, and (2) we separate planning into a two stage LLM-Modulo framework [30] with each stage designed to solve a different part of the overall planning problem. Samplers are named functions that take in one or more arguments, as well as particular arguments, and output a set of continuous values that may be useful for grounding skill(s). For instance, a simple uniform random sampler (which we call `Continuous` in the code and examples below) might sample a number uniformly at random within some provided bounds. A grasp sampler might take in no arguments and simply

---

[2]Note that our notion of constraint is broader than the typical notion in the CSP literature. Some of our constraints are *implicit*: they are checked via a simulator rather than expressed as simple symbolic expressions.

[3]Overall, our problem setting is analogous to that of TAMP [8] without symbolic predicates or operators.

output a valid grasp. Note importantly that these samplers are generally unaware of the constraints: a grasp sampler might output a grasp that is kinematically infeasible or unstable.

We leverage these samplers within a two-stage planning process. In the first stage of planning, which we call *LMP generation*, we prompt an LLM to generate a Language Model Program (LMP) [1]. This LMP is a function that takes in the text representation of the object-oriented state and certain parameters and outputs a plan that we assume achieves $\ell_G$. We also ask the LLM to generate bounds for and invoke the provided samplers to yield a sampling function that outputs values for the parameters of the LMP. In the next stage, which we call *constraint satisfaction*, we sample parameter choices for inputs to this LMP and execute them in our simulator to find parameters that ensure the plan does not violate any constraints from $C$. If the constraint satisfaction phase fails after a fixed sampling budget, we pass information about the most common constraints violated back to the LMP generation phase and request a new LMP and constraint bounds in light of the observed failure. An overview of this process is depicted in Figure 2.

We now discuss each phase in more detail. To ground this discussion, consider a simple running example in the Arrange-Blocks domain shown in Figure 4. Here, $\ell_G$ is: "Place the green block in the bowl". The state is represented using the `Object` class mentioned in Section 3, and the initial state is such that an orange block (`o12`) is atop the green block (`o7`). The robot is provided with a single simple continuous sampler and two skills, `pick(x,y,z)` and `place(x,y,z)`, that move the gripper to a particular `(x,y,z)` location and then close/open the gripper respectively.

**LMP Generation**: The objective of this stage is to generate an LMP that consists of: (1) a *plan-sketch function* that takes in a state as well as some arbitrary input parameters and outputs a plan $[\underline{\phi}_0, \underline{\phi}_1, ..., \underline{\phi}_K]$ when executed with an interpreter[4], and (2) a *sampling function* that leverages samplers with LLM-generated bounds to output parameters that (1) takes as input. Here, (1) together with the user-defined environment constraints $C$ defines a CCSP, and (2) helps define a sampling procedure that can be leveraged to solve this CCSP. To achieve this, we prompt an LLM with (a) the classes and objects used to represent the state-space $\mathcal{S}$, (b) the initial state $s_0$, (c) the available parameterized skills $\Phi$, (d) the provided samplers $\Sigma$, and (e) an example of an expected output LMP from a different task that shares the same state-space and many of the same skills (see Appendix B for the prompts used in our three environments). Importantly, note that the LLM is not provided with any of the constraints (rather, these will be checked in the next phase).

Consider the following generated LMP on our running example task of placing a block into a bowl:

```
def gen_plan(init:State, dx, dy):          def gen_domain(init:State):
    plan = []                                  return {
    block, bowl = init["o7"], init["o8"]          "dx": Continuous(-.04, .04),
    plan += [Action("pick", block.point)]         "dy": Continuous(-.04, .04),
    x, y, z = bowl.point                       }
    plan += [Action("place", [x+dx, y+dy, z])]
    return plan
```

Here, the `gen_plan` function is the plan-sketch. It takes in a particular state (named `init`) corresponding to the initial state $s_0$, as well as two parameters and generates a plan in terms of the provided `pick` and `place` skills. Importantly, note that the two input parameters to the function (namely `dx`, `dy`), are different from the parameters of the `pick` or `place` skills. The above generated sequence of one `pick` and one `place` skill would ordinarily require six continuous parameters (`x`, `y`, `z` for each skill) for grounding. However, the generated LMP reduces the sampling space to a lower-dimensional, two-parameter space (namely `dx`, `dy`). Thus, the generated LMP makes downstream CCSP simpler by leveraging the common-sense and reasoning capabilities of LLMs.

**Constraint Satisfaction and Feedback:** In this work, we opt for a very simple sample-and-test procedure for constraint satisfaction. Specifically, we sample a fixed number of values for all the plan-sketch's input variables (namely `dx` and `dy` in the above example) using the sampling function

---

[4]This function represents a *family* of plans that only differ in one or more continuous parameters.

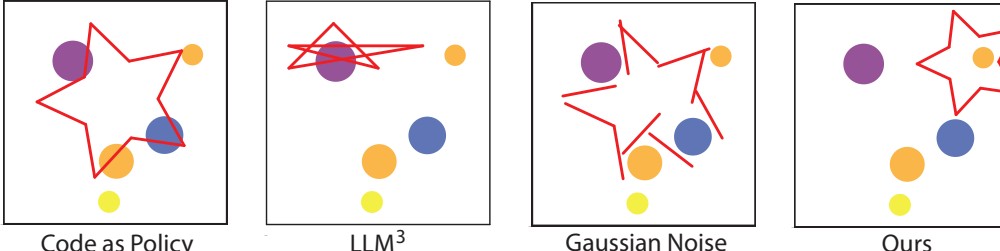

Figure 3: Top-down view of solutions produced by baselines on the star drawing task in our 'Drawing' domain.

that was generated by the previous stage. Given a particular sample, we can simply evaluate the LMP to output a plan. We then check for constraint violations by executing each step of this plan using our simulator $f$, and running each constraint classifier $c_i^\psi$ from $c_i \in C$.

If a plan is found that does not violate any constraints, we return this plan. If *no* satisfying plan is found after the fixed budget is exhausted, we enter a feedback stage. The objective of this stage is to provide information to the first stage such that it will return a new LMP that avoids the same constraint violations. In our implementation, we return the following information during feedback: (1) descriptions of the top 2 most common constraint violations, (2) the most common ground skill name that was run in the simulator before the violation, and (3) the most common index in the plan at which each of the above constraint violations occurred. This loop between LMP generation, constraint satisfaction and feedback continues until a legal plan is found.

In our running example, we exhaust our sampling budget due to collisions with the orange block and return the following feedback information: "Step 0, Action pick, Violation:Collision detected between object o12, gripper.". Given this, as well as the context of its previous LMP, the LLM generates a new LMP that moves object o12 out of the way before object o7 (the green block of interest) is manipulated:

```
def gen_plan(init:State, dx, dy,            def gen_domain(init:State):
        x_place_12, y_place_12):                return {
    plan = []                                    "dx": Continuous(-.04, .04),
    block_12 = init["o12"]                       "dy": Continuous(-.04, .04),
    plan += [Action("pick", block_12.point)]     "x_place_12": Continuous(
    plan += [Action("place", [x_place_12,                TABLE_BOUNDS[0][0],
        y_place_12, TABLE_BOUNDS[2][1]])]                TABLE_BOUNDS[0][1]),
    [block, bowl] = init["o7"], init["o8"]       "y_place_12": Continuous(
    plan += [Action("pick", block.point)]                TABLE_BOUNDS[1][0],
    x, y, z = bowl.pose.point                            TABLE_BOUNDS[1][1])
    plan += [Action("place", [x+dx, y+dy, z])]   }
    return plan
```

## 5 Experiments

Our experiments are designed to test the ability of our method (PRoC3S) to sequence a set of simple continuously parameterized skills to generalize to satisfying unseen natural language goals while obeying environmental constraints, both in simulated domains and on real-world hardware.

**Constraints.** We make use of four general constraint types across environments. These are *kinematic constraints* on the robot, *collision constraints* for robot motion, *grasp constraints* (i.e., checking for stable grasps), and *placement constraints* (i.e., checking for stable placements). Details on implementation are provided in Appendix A.2.

**Environments.** We now provide high-level environment and task descriptions with details in Appendix A. All of our simulated domains consist of a 6 DoF UR5 robot arm with a Robotiq 2F-85 gripper in front of a table of objects. Experiments in these domains involve different initial states,

| | Drawing | | | | Arrange Blocks | | | | Arrange YCB | |
|---|---|---|---|---|---|---|---|---|---|---|
| | Star | Arrow | Letters | Enclosed | Pyramid | Line | Packing | Unstack | Packing | Stacking |
| LLM³ | 40% | 40% | **90%** | 50% | 0% | **50%** | 30% | 20% | 0% | 0% |
| LLM³-NF | 20% | 0% | 40% | 20% | 10% | 30% | **60%** | 20% | 0% | 0% |
| LLM³-Gaussian | 20% | 0% | 0% | 0% | **30%** | 40% | 30% | 20% | 0% | 0% |
| CaP | 10% | 0% | 50% | 30% | 20% | 20% | 20% | 20% | **40%** | 10% |
| CaP-Gaussian | 10% | 20% | 0% | 40% | 10% | 30% | 30% | 30% | 20% | 10% |
| PRoC3S-NF | **100%** | 40% | 50% | **90%** | 30% | 10% | **70%** | 20% | 10% | **40%** |
| PRoC3S | **80%** | **80%** | 80% | **90%** | **60%** | **70%** | **70%** | **70%** | **60%** | **70%** |

Table 1: Percentage of correct final states over 10 evaluations across simulated domains. Values not significantly different from the top performer are bolded. Tests are a one-tailed Z-test with $\alpha = 0.1$.

object sets, and goals based on the simulated Ravens tabletop environment first introduced by Zeng et al. [31]. Our real-world domain consists of a tabletop setup with a Franka Emika Panda robot.

- *Drawing*: The robot is provided with a parameterized skill to draw a line, and a variety of goals involving drawing different shapes. We attempt four different goals in this environment: "draw a star", "draw an arrow pointing at the biggest obstacle in the environment", "draw the letter M", and "draw a shape that encloses two obstacles". The main challenge of these tasks is drawing a shape that avoids collisions with the obstacles in the cluttered environment.
- *Arrange-Blocks*: A tabletop in front of the robot is strewn with a variety of colored blocks and bowls. We attempt 4 tasks in this environment: "stack an upright pyramid out of three blocks", "Put five blocks in a line flat on the table", "Place all blocks within 0.06 of the center of the table", and "Place the green block in a bowl". The first three are challenging due to low-tolerance stability and collision constraints. The final task is challenging because there are always other blocks atop the green one that prevent it from being picked directly.
- *Arrange-YCB*: The same as the above Arrange-Blocks environment, but with objects from the YCB dataset [32] instead of blocks and bowls. The goals are: "Place all objects within 0.06 of the center of the table", "stack any two objects". Both of these tasks require satisfying grasp constraints on arbitrary object meshes and kinematic/reachability constraints on the selected grasps. In the case of the packing problem, collision constraints between objects are the main constraint violation, whereas placement stability is more significant for the stacking task.
- *Real Robot Domain*: We implement 4 tasks from the 'Arrange-Blocks' and 'Arrange-YCB' domains on a real-world tabletop setup with a Franka Emika Panda robot. The goals are: "Put three/five blocks in a line flat on the table" (referred to as '3-line' and '5-line' respectively), "Place all blue blocks in the blue bowl and red blocks in the red bowl" ('Sort'), and "Place all objects within 0.06 of the center of the table" ('YCB-packing'). See Appendix D for an illustration of our setup, as well as implementation details necessary to apply our approach to this challenging domain. Note that we report results from directly executing computed plans 'open-loop' on the real robot and do not attempt to replan upon failure.

**Approaches.** We now briefly describe the approaches that we compare to PRoC3S.

- *PRoC3S without feedback (PRoC3S-NF)*: PRoC3S but with the feedback component ablated. Thus, if the first returned plan doesn't work, we consider the task failed.
- *Code as Policies (CaP)* [1]: This approach attempts to write helper functions and leverage existing Python libraries to output an LMP that produces skill sequences and corresponding continuous parameters give a task, without any environment feedback.
- *CaP-Gaussian*: In an approach loosely based on a previous paper [18], We add gaussian noise to the action space of the skills output by the LMP generated by *CaP*, which can help avoid constraints violations but detracts from the intended goal. See Appendix B.5 for details.
- *LLM³*[3]:. This approach performs LLM planning with parameterized skills and feedback, but uses the LLM to directly output continuous parameters. We also ablate the feedback component of this approach (*LLM³-NF*), and include a version that places Gaussian noise on the output similar to CaP-Gaussian (*LLM³-Gaussian*).

**Experimental Setup.** For each task and approach, we run 10 random seeds where we randomize the initial locations and sizes (where appropriate) of objects. For our real robot domain, we run 5 random

seeds of the approach that had the best performance across simulated domains with randomization of the initial state. For approaches that use feedback (i.e., PRoC3S and LLM$^3$), we limit the number of feedback iterations to 5. We specify three samplers ($\Sigma$) and provide all approaches with access to them (details in Appendix C and Appendix B). We fix a sampling budget of 10000 for tasks in the Drawing domain, and 1000 for all other domains. For all approaches, we use the OpenAI GPT-4 LLM [33], specifically the gpt-4-0125-preview checkpoint. We record success on achieving the natural language task goal as judged by the authors and report additional metrics in Appendix A.1.

**Results and Analysis.** As can be seen from Table 1, PRoC3S consistently achieves the highest success rate across simulated domains. PRoC3S-NF's success rate is at least 30% lower on most tasks, illustrating the importance of feedback. This is further validated by the fact that the performance of LLM$^3$-NF is significantly worse than LLM$^3$. We see that LLM$^3$ itself performs comparably to PRoC3S on 2 tasks, and find that it falters because the LLM is unable to directly output continuous parameters that satisfy the various task constraints, as can be seen from the example in Figure 5. CaP is only comparable to our approach on one task and generally fails because it is unable to generate viable continuous parameters (e.g. Figure 5). While gaussian noise helps these baselines avoid constraint violations, it prevents them from achieving the overall task goal as shown in Figure 5. Additionally, Table 2 indicates that our approach is able to operate on challenging real-world variants of several simulated tasks and achieve a non-trivial success rate.

Qualitatively, we observe two common failures in simulation. Firstly, we observe cases in the 'Arrange Blocks' and 'Arrange YCB' domains where our method cannot correctly refine its LMP given feedback. Once an incorrect modification is made, we

| Seed | 3-line | 5-line | Sort | YCB-packing |
|------|--------|--------|------|-------------|
| 1 | ✓ | ✓ | ✓ | ✓ |
| 2 | ✓ | Grasp fail | ✓ | ✓ |
| 3 | ✓ | CSP timeout | ✓ | Grasp fail |
| 4 | ✓ | ✓ | ✓ | Grasp fail |
| 5 | ✓ | ✓ | Self-collision | Object collision |

Table 2: Results on real-world robot tasks. ✓ indicates success.

find that the LLM rarely recovers in future feedback iterations. Secondly, as observed in prior works [34, 30], we find that the LLM is unable to yield a valid plan for the longer horizon problems. On the real-world tasks, we observe that most failures stem from the sim-to-real gap. Only one observed failure ('5-line' task seed 3) was due to our method failing to find a plan: all other failures were due to failed grasps or unexpected collisions during real-world execution after planning.

## 6 Limitations and Future Work

There are several limitations of our method. Firstly, our method requires a physics simulator, which introduces a significant sim-to-real gap that can lead to execution failures (as observed in Table 1). Secondly, the open parameters in the generated LMPs from the first phase of our method depend heavily on the example task we choose to provide as part of the input prompt, making effective prompting crucial for the success of our approach. Thirdly, our method of solving CCSPs is naive and can be very slow, especially in domains like Arrange-YCB. Additionally, our framework guarantees neither soundness nor completeness nor optimality of the generated plans.

In future work, we aim to improve the sampling technique for continuous parameters, opting for a backtracking search or optimization algorithm to select parameter values as part of the sampling phase of our method. As part of this, we hope to explore expressing constraints in terms of continuous cost functions instead of binary failure detectors. We also plan to instill the robot with visual reasoning by exploring the use of vision language models (VLMs), potentially enabling it to recognize a wide range of constraint violations without us having to explicitly define constraint classifier functions. Lastly, we hope to extend our method to partially-observable environments and thereby tackle larger mobile-manipulation tasks.

## Acknowledgements

We gratefully acknowledge support from NSF grant 2214177; from AFOSR grant FA9550-22-1-0249; from ONR MURI grant N00014-22-1-2740; from ARO grant W911NF-23-1-0034; from the MIT Quest for Intelligence; and from the Boston Dynamics Artificial Intelligence Institute. Aidan and Nishanth are supported by NSF Graduate Research Fellowships. Any opinions, findings, and conclusions or recommendations expressed in this material are those of the authors and do not necessarily reflect the views of our sponsors.

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

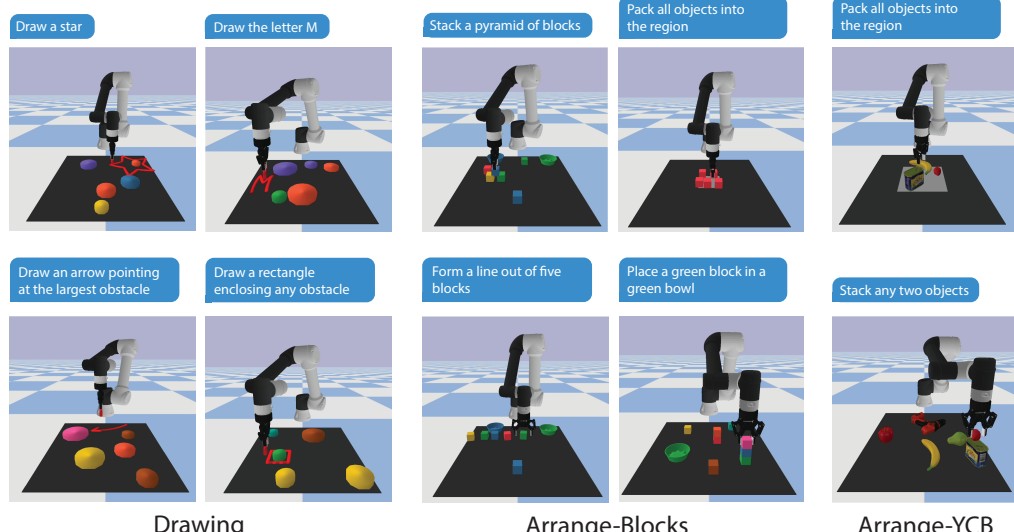

Figure 4: Illustration of tasks in our simulated environments, along with corresponding language goals.

# A    Simulated Environment Details and Setup

In this section, we describe details on the constraints implemented across environments, as well as the setup for each task in each environment shown in Figure 4. All experiments in simulated domains were conducted on machines with 8-core CPU's and 32GB of RAM.

## A.1    Additional Experiment Statistics

Tables 3 and 4 show the number of samples and wall-clock time required by our various approaches to perform constraint satisfaction. As can be seen from these results, performing sampling is critical to solving each of our presented tasks. PRoC3S generally requires the smallest number of CSP samples compared to baselines, especially in the first two tasks of the 'Drawing' domain. For tasks where our approach requires more CSP samples (e.g. 'Pyramid' and 'Unstack' in 'Arrange Blocks', and 'Packing' in 'Arrange YCB'), our approach achieves a much higher success rate, indicating the baselines were able to terminate easily because they yielded incorrect plan generators. We observe a similar trend in Table 5, which shows the overall planning time required by each method. Table 6 lists the number of feedback queries (i.e., number of times the LLM was queried for an LMP after the first time) across the various baselines that use feedback. Compared to LLM[3], our approach uses significantly fewer feedback iterations on all tasks except 'Unstack', 'Packing', and 'Stacking' in which our approach has significantly higher success rate. This is especially noteworthy in the 'Drawing' domain, where our approach requires less than 1 feedback iteration on average. Finally, Table 7 lists the wall-clock time spent querying the LLM for all approaches and tasks. We see that a significant fraction of the total planning time taken by our approach (over $50\%$ in most tasks) is spent querying the LLM. There are only relatively few tasks (e.g. 'Packing' and 'Unstack' in the 'Arrange Blocks' domain) where sampling takes up the majority of the planning time.

## A.2    Constraints

- *Kinematic Constraints*: We make use of the PyBullet [35] inverse kinematic solver to reach desired end-effector positions. If the solver returns joint positions that result in an incorrect end-effector position, we deem the target as kinematically infeasible.
- *Collision Constraints*: If the gripper collides with any unexpected objects during robot motion, we consider this a collision constraint violation. Expected collisions, such as those between the gripper and held object, are not considered collisions.

|  | Drawing | | | | Arrange Blocks | | | | Arrange YCB | |
|---|---|---|---|---|---|---|---|---|---|---|
|  | Star | Arrow | Letters | Enclosed | Pyramid | Line | Packing | Unstack | Packing | Stacking |
| LLM³ | - | - | - | - | - | - | - | - | - | - |
| LLM³-NF | - | - | - | - | - | - | - | - | - | - |
| LLM³-Gaussian | 1983.10± 3288.05 | 1371.80± 3028.34 | 1148.40± 724.10 | 5047.50± 4558.19 | 274.00± 451.81 | 897.00± 433.77 | 1555.67± 1422.02 | 176.80± 192.37 | 0.00± 0.00 | 0.00± 0.00 |
| CaP | - | - | - | - | - | - | - | - | - | - |
| CaP-Gaussian | 1799.00± 2834.45 | 1145.90± 2959.80 | 2318.90± 2733.69 | 4172.10± 4766.29 | 174.40± 113.95 | 199.20± 99.60 | 126.00± 329.98 | 134.40± 115.76 | 899.10± 299.70 | 249.00± 0.00 |
| PRoC3S-NF | 5.70± 9.61 | 8.60± 10.01 | 2001.50± 3998.75 | 1001.20± 2999.27 | 39.80± 74.34 | 52.40± 88.91 | 513.78± 462.69 | 199.20± 99.60 | 40.80± 116.52 | 8.56± 11.31 |
| PRoC3S | 3.50± 5.68 | 8.70± 10.13 | 1000.70± 3000.10 | 5000.80± 14998.07 | 336.60± 456.99 | 54.50± 148.30 | 312.22± 403.32 | 523.70± 465.13 | 1160.00± 1332.93 | 3.30± 2.24 |

Table 3: Average number of samples required to solve CSP. Standard deviation is indicated after ±. Baselines that do not leverage sampling to solve the CSP are listed with a −. Note that these results do not enable direct comparison of methods (i.e., lower numbers are not necessarily better) because they do not account for success rate from Table 1.

|  | Drawing | | | | Arrange Blocks | | | | Arrange YCB | |
|---|---|---|---|---|---|---|---|---|---|---|
|  | Star | Arrow | Letters | Enclosed | Pyramid | Line | Packing | Unstack | Packing | Stacking |
| LLM³ | - | - | - | - | - | - | - | - | - | - |
| LLM³-NF | - | - | - | - | - | - | - | - | - | - |
| LLM³-Gaussian | 3.64± 5.25 | 2.35± 4.81 | 1.85± 1.17 | 8.49± 7.80 | 2.54± 3.99 | 19.28± 16.41 | 18.27± 16.01 | 0.90± 1.16 | 0.00± 0.00 | 0.00± 0.00 |
| CaP | - | - | - | - | - | - | - | - | - | - |
| CaP-Gaussian | 4.02± 6.26 | 1.93± 4.63 | 3.58± 4.26 | 6.63± 7.52 | 2.67± 1.52 | 3.42± 2.55 | 2.75± 5.36 | 0.68± 0.36 | 473.53± 272.29 | 330.15± 220.86 |
| PRoC3S-NF | 0.02± 0.02 | 0.02± 0.02 | 2.69± 5.36 | 1.26± 3.77 | 6.33± 8.22 | 23.74± 50.77 | 193.76± 222.47 | 1.13± 0.50 | 30.50± 86.47 | 6.34± 5.49 |
| PRoC3S | 0.02± 0.01 | 0.02± 0.02 | 1.26± 3.75 | 10.70± 32.06 | 79.69± 132.53 | 4.98± 4.93 | 165.51± 230.33 | 29.13± 53.43 | 648.26± 716.99 | 8.05± 13.09 |

Table 4: Average wall clock time spent on continuous constraint satisfaction. Standard deviation is indicated after ±. Baselines that do not use constraint satisfaction are listed with a −. Note that these results do not enable direct comparison of methods (i.e., lower numbers are not necessarily better) because they do not account for success rate from Table 1.

- *Grasp Constraints*: We consider the following as constraint violations when selecting grasps: (1) the end-effector pose (before the gripper is closed) is in collision with the robot or any other objects in the scene, (2) no part of the object is between the two gripper fingers, (3) the object falls out of the hand when tested in simulation.
- *Placement Constraints*: If the object moves significantly if we step physics simulation for a fixed number of time steps after the gripper opens to release it, we consider this a placement constraint violation.

## A.3  Drawing

For each of the four drawing tasks, we randomize the position of five circles with random radii.

### A.3.1  Skills

`draw_line(p1_x, p1_y, p2_x, p2_y)`
Moves the arm to a point (`p1_x, p1_y, 0.1`) (where the table surface is at height 0.0), draws a straight line from point (`p1_x, p1_y`) to point (`p2_x, p2_y`) while keeping the pen at $z$ pose 0.0, and then moves the arm back up to the point (`p1_x, p1_y, 0.1`). The orientation of the gripper is fixed to be top-down throughout.

| | Drawing | | | | Arrange Blocks | | | | Arrange YCB | |
|---|---|---|---|---|---|---|---|---|---|---|
| | Star | Arrow | Letters | Enclosed | Pyramid | Line | Packing | Unstack | Packing | Stacking |
| LLM³ | 29.11± 15.78 | 14.62± 7.44 | 20.48± 11.42 | 28.28± 5.69 | 16.04± 12.95 | 45.25± 14.82 | 24.41± 11.31 | 4.33± 0.84 | 0.00± 0.00 | 0.00± 0.00 |
| LLM³-NF | 7.45± 2.48 | 5.21± 1.14 | 5.47± 1.16 | 5.18± 1.29 | 5.07± 0.86 | 8.38± 1.26 | 6.48± 0.85 | 2.23± 0.26 | 0.00± 0.00 | 6.63± 0.94 |
| LLM³-Gaussian | 11.46± 7.53 | 9.09± 7.91 | 7.40± 1.43 | 16.19± 10.69 | 13.25± 11.58 | 86.09± 54.32 | 37.61± 26.14 | 7.15± 7.25 | 0.00± 0.00 | 0.00± 0.00 |
| CaP | 15.01± 1.66 | 10.36± 1.43 | 16.82± 2.16 | 17.02± 2.39 | 15.77± 2.64 | 16.57± 4.66 | 15.17± 4.07 | 12.34± 14.50 | 15.88± 3.09 | 15.57± 2.67 |
| CaP-Gaussian | 19.71± 8.85 | 11.95± 4.70 | 20.66± 4.11 | 23.53± 10.30 | 15.68± 2.85 | 22.87± 16.39 | 17.05± 5.79 | 13.33± 14.53 | 489.65± 273.18 | 349.08± 220.48 |
| PRoC3S-NF | 19.34± 2.44 | 20.85± 3.82 | 26.10± 6.57 | 18.26± 5.65 | 42.01± 35.99 | 42.47± 51.83 | 213.86± 220.92 | 14.51± 1.67 | 53.31± 87.38 | 25.51± 7.35 |
| PRoC3S | 20.50± 3.55 | 20.35± 2.68 | 30.41± 10.15 | 37.93± 58.66 | 140.48± 172.30 | 44.62± 38.85 | 187.65± 236.75 | 121.95± 130.72 | 727.57± 751.57 | 32.28± 15.75 |

Table 5: Average wall-clock time taken to return a plan (i.e., perform LMP generation and sampling to find a non-violating solution, including any feedback iterations). Standard deviation is shown after ±. Note that these results do not enable direct comparison of methods (i.e., lower numbers are not necessarily better) because they do not account for success rate from Table 1.

| | Drawing | | | | Arrange Blocks | | | | Arrange YCB | |
|---|---|---|---|---|---|---|---|---|---|---|
| | Star | Arrow | Letters | Enclosed | Pyramid | Line | Packing | Unstack | Packing | Stacking |
| LLM³ | 3.10± 1.97 | 2.10± 1.64 | 2.60± 1.80 | 4.00± 0.89 | 1.90± 2.21 | 4.00± 1.49 | 2.44± 1.71 | 0.70± 0.46 | 0.00± 0.00 | 0.00± 0.00 |
| LLM³-NF | - | - | - | - | - | - | - | - | - | - |
| LLM³-Gaussian | 0.10± 0.30 | 0.10± 0.30 | 0.00± 0.00 | 0.40± 0.49 | 1.10± 1.81 | 3.70± 1.79 | 1.56± 1.42 | 0.70± 0.78 | 0.00± 0.00 | 0.00± 0.00 |
| CaP | - | - | - | - | - | - | - | - | - | - |
| CaP-Gaussian | - | - | - | - | - | - | - | - | - | - |
| PRoC3S-NF | - | - | - | - | - | - | - | - | - | - |
| PRoC3S | 0.00± 0.00 | 0.00± 0.00 | 0.10± 0.30 | 0.50± 1.50 | 1.50± 2.11 | 0.20± 0.60 | 0.22± 0.42 | 2.10± 1.81 | 2.20± 2.04 | 0.30± 0.64 |

Table 6: Average number of times feedback was queried. Standard deviation is indicated after ±. Approaches that do not leverage feedback are listed with a −. Note that these results do not enable direct comparison of methods (i.e., lower numbers are not necessarily better) because they do not account for success rate from Table 1.

## A.4 Arrange-Blocks

For the pyramid-stacking task and the line-forming task, we randomize the position of two bowls and six blocks such that no objects are stacked on top of each other.

For the region-packing task, we randomize the position of five red blocks and create a low square prism centered at the middle of the table to represent the region for the red blocks to be packed. The red blocks are arranged so that no blocks are stacked on top of each other.

For the task of placing a green block into a green bowl, we randomize the position of a green bowl and eight blocks with at least one green block. Blocks may be stacked on top of each other.

### A.4.1 Skills

`pick(x, y, z)`
Move the gripper to location (`x, y, z`) and close the gripper.
The `pick` skill moves the robot's gripper to a specified location (`pick_pose`), closes the gripper to grasp an object, and then lifts the object back to a hover position that is simply (x, y, z + 0.1) (where the table height is 0.0). The orientation of the gripper is fixed to be top-down throughout.

`place(x, y, z)`
Move the gripper to location (`x, y, z`) and open the gripper.

| | Drawing | | | | Arrange Blocks | | | | Arrange YCB | |
|---|---|---|---|---|---|---|---|---|---|---|
| | Star | Arrow | Letters | Enclosed | Pyramid | Line | Packing | Unstack | Packing | Stacking |
| LLM³ | 29.11± 15.78 | 14.62± 7.44 | 20.48± 11.42 | 28.28± 5.69 | 16.04± 12.95 | 45.25± 14.82 | 24.41± 11.31 | 4.33± 0.84 | 0.00± 0.00 | 0.00± 0.00 |
| LLM³-NF | 7.45± 2.48 | 5.21± 1.14 | 5.47± 1.16 | 5.18± 1.29 | 5.07± 0.86 | 8.38± 1.26 | 6.48± 0.85 | 2.23± 0.26 | 0.00± 0.00 | 6.63± 0.94 |
| LLM³-Gaussian | 7.82± 5.05 | 6.74± 4.60 | 5.55± 0.90 | 7.69± 3.17 | 10.70± 7.65 | 66.81± 50.45 | 19.34± 10.32 | 6.26± 6.19 | 0.00± 0.00 | 0.00± 0.00 |
| CaP | 15.01± 1.66 | 10.36± 1.43 | 16.82± 2.16 | 17.02± 2.39 | 15.77± 2.64 | 16.57± 4.66 | 15.17± 4.07 | 12.34± 14.50 | 15.88± 3.09 | 15.57± 2.67 |
| CaP-Gaussian | 15.69± 3.76 | 10.02± 2.96 | 17.08± 3.68 | 16.89± 3.92 | 13.01± 2.48 | 19.45± 14.52 | 14.30± 3.30 | 12.65± 14.38 | 16.12± 2.54 | 18.93± 2.27 |
| PRoC3S-NF | 19.33± 2.44 | 20.83± 3.82 | 23.42± 4.60 | 17.00± 2.71 | 35.68± 33.70 | 18.73± 4.27 | 20.10± 3.32 | 13.38± 1.81 | 22.82± 3.55 | 19.16± 3.18 |
| PRoC3S | 20.49± 3.55 | 20.33± 2.68 | 29.16± 6.64 | 27.23± 26.62 | 60.79± 54.54 | 39.64± 38.97 | 22.14± 7.28 | 92.82± 83.34 | 79.31± 56.18 | 24.23± 12.78 |

Table 7: Average wall-clock time spent querying the language model in seconds. Standard deviation is included after ±. Note that these results do not enable direct comparison of methods (i.e., lower numbers are not necessarily better) because they do not account for success rate from Table 1.

The `place` skill moves the robot's gripper to a specified location (`place_pose`) to place the object and releases the object. It then moves the gripper back to a hover position that is simply (x, y, z + 0.1) (where the table height is 0.0). The orientation of the gripper is fixed to be top-down throughout.

### A.5 Arrange-YCB

For the region-packing task, we randomize the positions of three objects: a banana, a strawberry, and a meat can. We also create a low square prism centered on the table to represent the packing region for these objects.

For the stacking task, we randomize the positions of six objects: a banana, a power drill, a meat can, a strawberry, an apple, and a pear.

#### A.5.1 Skills

`pick(o, g)`
Pick up object o at grasp g sampled from a grasp sampler.
The `pick` skill moves the robot's gripper to a target object's position (`gripper_target`) using predefined poses (`hover_pose` and `gripper_target`). The gripper closes around the object and lifts the object back to the hover position.

`place(o, g, p)`
If holding an object o at grasp g, place the object at pose p.
The `place` skill moves the robot's gripper to a target location (`gripper_target`) using a hover position. It then releases the object and the robot moves back to the hover position.

## B  PRoC3S Prompting Details

Here we provide details on the prompting scheme used for each environment. As outlined in Section 4, the initial prompt to the LLM consists of (a) the classes and objects used to represent the state-space $\mathcal{S}$, (b) the initial state $s_0$, (c) the available parameterized skills $\Phi$, and (d) the provided samplers $\Sigma$, and (e) an example of an expected output from a different task. We now provide the prompts we use for each of our environments and task. We start by providing a common prompt 'template' that's shared by all tasks in an environment. We then further specify elements that differ between tasks. Since there exists a domain example for each domain/method combination, we point readers to our public code release for full example prompts.

The prompting template for each environment is structured as follows:

{{{system_prompt}}}

{{{domain_setup_code}}}

{{{skill_preface}}}

{{{domain_skills}}}

{{{method_role}}}

{{domain_example}}

The `system_prompt`, `skill_preface`, and `role` are identical for all three environments and establish context for the robot.

`system_prompt`:

```
#define system
You are a robot operating in an environment
with the following state
```

`skill_preface`:

```
You have access to the following set of skills expressed as pddl predicates followed
↪  by descriptions.
You have no other skills you can use, and you must exactly follow the number of
↪  inputs described below.
The coordinate axes are x, y, z where x is distance from the robot base, y is
↪  left/right from the robot base, and z is the height off the table.
```

We now provide prompts which are unique to each environment: `domain_setup_code`, `domain_skills`, example input task details, and example LMP output.

## B.1 Drawing

`drawing_setup_code`

```python
COLORS = ["blue", "green", "pink", "purple"]

@dataclass
class Obstacle:
    name: str
    x_pos: float
    y_pos: float
    radius: float
    color: str

@dataclass
class DrawnLine:
    p1_x: float
    p1_y: float
    p2_x: float
    p2_y: float

@dataclass
class DrawingState:
    obstacles:List[Obstacle] = field(default_factory=list)
    drawn_lines:List[DrawnLine] = field(default_factory=list)

@dataclass
class ContinuousSampler:
    min: float = 0
```

```
        max: float = 1

        def sample(self):
            return random.uniform(self.min, self.max)

@dataclass
class DiscreteSampler:
    values: List[int]

    def sample(self):
        return random.choice(self.values)

@dataclass
class Action:
    name: str
    params: List[float]
```

drawing_skills

```
Action("draw_line", [p1_x, p1_y, p2_x, p2_y])
Draws a straight line from (p1_x, p1_y) to (p2_x, p2_y).
The pen is lifted up to get to the start of the next action.
```

## B.2 Arrange

arrange_setup_code

```
CATEGORIES = ["bowl", "block"]
TABLE_BOUNDS = [[-0.3, 0.3], [-0.8, -0.2], [0, 0]]  # X Y Z
TABLE_CENTER = [0, -0.5, 0]
BLOCK_SIZE = 0.04

@dataclass
class ArrangePose:
    x: float = 0
    y: float = 0
    z: float = 0
    roll: float = 0
    pitch: float = 0
    yaw: float = 0

    @property
    def point(self):
        ...

    @property
    def euler(self):
        ...

@dataclass
class ArrangeObject:
    category: str
    color: str
    pose: ArrangePose = field(default_factory=lambda: ArrangePose())
    body: Optional[int] = None

@dataclass
class ArrangeBelief:
    objects: Dict[str, ArrangeObject] = field(default_factory=dict)
    observations: List[Any] = field(default_factory=list)

@dataclass
class ContinuousSampler:
    min: float = 0
```

```
    max: float = 1

    def sample(self):
        return random.uniform(self.min, self.max)

@dataclass
class Action:
    name: str
    params: List[float]
```

arrange_skills

```
Action("pick", [x, y, z])
Move to the gripper to location x, y, z and close the gripper

Action("place", [x, y, z])
Move to the gripper to location x, y, z and open the gripper
```

## B.3   Arrange YCB

arrange_ycb_setup_code

```
CATEGORIES = ["bowl", "block"]
TABLE_BOUNDS = [[-0.3, 0.3], [-0.8, -0.2], [0, 0]]  # X Y Z
TABLE_CENTER = [0, -0.5, 0]
BLOCK_SIZE = 0.04

@dataclass
class ArrangePose:
    x: float = 0
    y: float = 0
    z: float = 0
    roll: float = 0
    pitch: float = 0
    yaw: float = 0

    @property
    def point(self):
        ...

    @property
    def euler(self):
        ...

@dataclass
class ArrangeObject:
    category: str
    color: str
    pose: ArrangePose = field(default_factory=lambda: ArrangePose())
    body: Optional[int] = None

@dataclass
class ArrangeBelief:
    objects: Dict[str, ArrangeObject] = field(default_factory=dict)
    observations: List[Any] = field(default_factory=list)

@dataclass
class ContinuousSampler:
    min: float = 0
    max: float = 1

    def sample(self):
        return random.uniform(self.min, self.max)
```

```python
@dataclass
class DiscreteSampler:
    values: List[int]

    def sample(self):
        return random.choice(self.values)

@dataclass
class Action:
    name: str
    params: List[float]

@dataclass
class GraspSampler(Sampler):
    def sample(self) -> ArrangeGrasp:
        ...
```

arrange_ycb_skills

```
Action("pick", [o, g])
Pick up object o at grasp g sampled from a grasp sampler. Grasps MUST come from
↪   grasp samplers.

Action("place", [o, g, p])
If holding an object o at grasp g, place the object at pose p.
```

## B.4   Method Prompts

We now go through all of the method specific prompts, which involve a role specification and a method-specific example.

proc3s_role:

```
Your goal is to generate two things:

First, generate a python function named `gen_plan` that can take any discrete or
↪   continuous inputs. No list inputs are allowed.
and return the entire plan with all steps included where the parameters to the plan
↪   depend on the inputs.

Second, generate a python function `gen_domain` that returns a set of bounds for the
↪   continuous or discrete input parameters. The number of bounds in the
generated domain should exactly match the number of inputs to the function excluding
↪   the state input

The function you give should always achieve the goal regardless of what parameters
↪   from the domain are passed as input.
The `gen_plan` function therefore defines a family of solutions to the problem.
↪   Explain why the function will always satisfy the goal regardless of the input
↪   parameters.
Make sure your function inputs allow for as much variability in output plan as
↪   possible while still achieving the goal.
Your function should be as general as possible such that any correct answer
↪   corresponds to some input parameters to the function.

All of these parameter samples may fail, in which case it will return feedback about
↪   what constraints caused the failure.
In the event of a constraint satisfaction fail, explain what went wrong and then
↪   return an updated gen_plan and gen_domain that fixes the issue.

This may involve adding actions to the beginning of the plan to move obstructing
↪   objects leading to collisions and adding new continuous input parameters that
↪   are used for those new actions.
```

```
Do not add complex logic or too much extra code to fix issues due to constraint
↪   violations.

The main function should be named EXACTLY `gen_plan` and the domain of the main
↪   function should be named EXACTLY `gen_domain`. Do not change the names. Do not
↪   create any additional classes or overwrite any existing ones.
Aside from the inital state all inputs to the `gen_plan` function MUST NOT be of
↪   type List or Dict. List and Dict inputs to `gen_plan` are not allowed.

cap_role:

Your goal is to generate a python function that returns a plan that performs the
↪   provided task. This function can
use helper functions that must be defined within the scope of the function itself.

The main function should be named EXACTLY `gen_plan`, and it should take in only one
↪   parameter corresponding to the environment state as input. Do not change the
↪   names. Do not create any additional classes or overwrite any existing ones. You
↪   are only allowed to create helper functions inside the `gen_plan` function.

llm3_role:

Your goal is to generate a sequence of actions that are stored in a variable named
↪   `gen_plan`

The checker may fail, in which case it will return feedback about what constraints
↪   caused the failure.
In the event of a failure, propose a modified plan that avoids all potential reasons
↪   for failure.

DO NOT use placeholders, equations, mathematical operations.
Always give a ground plan that could be directly executed in the environment.

You must always return a block of python code that assigns a list of actions to a
↪   variable named EXACTLY `gen_plan`
```

## B.5    Gaussian Sampling

Gaussian sampling is applied to the parameter space of the skills outputted by the action, which means it can be used for any method. The sampling uses a similar feedback loop to constraint satisfaction and runs for the same number of samples as constraint satisfaction (domain dependent). The initial Gaussian standard deviation is set to zero and is increased linearly to 1 until either a solution is found with no constraint violations or the maximum number of samples is reached.

# C    Sampler Details

We implement 3 different samplers ($\Sigma$) in python:

```python
@dataclass
class ContinuousSampler(Sampler):
    min: float = 0
    max: float = 1
    def sample(self):
        return random.uniform(self.min,
                              self.max)

@dataclass
class DiscreteSampler:
    values: List[int]
    def sample(self):
        return random.choice(self.values)

@dataclass
```

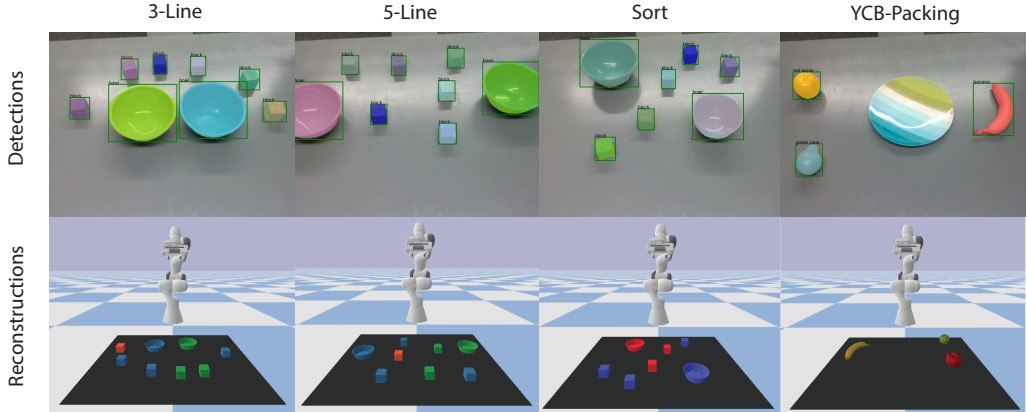

Figure 5: Examples from the perception system used as part of implementing our real-world domains. See Appendix D for details.

```python
class GraspSampler(Sampler):
    def sample(self) -> ArrangeGrasp:
        return ArrangePose(
            x=random.uniform(-0.02, 0.02),
            y=random.uniform(-0.02, 0.02),
            pitch=np.pi,
            yaw=random.uniform(-math.pi,
                               math.pi),
        ).multiply(ArrangePose(z=-0.005))
```

The `GraspSampler` is only provided in the Arrange-YCB environment.

## D    Real-world Robot Implementation Details

Similar to TAMP approaches [36, 37], we deploy PRoC3S on real-world hardware by: (1) leveraging pre-trained vision models to estimate the features of some known set of objects in a real-scene, (2) instantiating a digital-twin simulation in which we perform planning, (3) executing the plan open-loop on the real-world robot [5].

Our implementation assumes access to (1) a set of object models compatible with a physics simulator, and (2) a set of strings describing each object model (e.g. "apple"). Each of these models and its associated string corresponds to an object class. Given these, we leverage open-vocabulary object detection and segmentation (specifically, Ren et al. [38]) to segment and associate portions of a point-cloud with each string. For each detection, we extract the centroid of the segment, transform this into the robot's coordinate system, and use these as the position (i.e., $x, y, z$) of the corresponding object model within our physics simulator. While full pose-estimation is possible within this setup, we leave this for future work. Additionally, we extract object colors from the masked point-cloud returned by object detection. This is done by taking the average RGB values of all points in the pointcloud, and retrieving the closest color (as determined by L2 distance) to a predefined color set. Each object is thus equipped with a feature called 'color' with a string value referring to the name of its color. While assuming object models can be a strong assumption, model-free versions of this setup have been demonstrated to work on TAMP systems [37], and could similarly be adopted in this framework.

---

[5] If the real-world execution fails, then the system can be simply made to 'replan' by restarting from step (1)

