# OpenReview forum: "Trust the PRoC3S: Solving Long-Horizon Robotics Problems with LLMs and Constraint Satisfaction"
_robot-learning.org/CoRL/2024/Conference — CoRL 2024_

### Official Review · Reviewer_XQei · 2024-07-19
**Review of Trust the PRoC3S**

**Originality:** 3
**Technical Quality:** 3
**Clarity Of Presentation:** 4
**Potential Impact:** 2
**Recommendation:** 2
**Confidence:** 4

**Review:**

PRoC3S is a method for augmenting LLM planning with constraint satisfaction. The authors do so by integrating samplers for continuously parameterized skills into the planning process along with a simulator, enabling the LLM to plan, test and iterate till it solves the task with the given constraints. The results demonstrate clear improvement over baseline methods that do not leverage this information and are unable to take into account constraints.

Strengths:
* well written paper that clearly communicates the ideas and method
* proposed method improves results over baselines across a range of tasks

Weaknesses:
* Largely unclear how this method can be used for real robots due to its significant assumptions about privileged information
* Missing comparisons to relevant LLM + task planning baselines such as LLM+P (see questions)
*  Analysis is a bit lacking, no experiments done beyond main results (see questions)
* Motivation is a bit shaky: TAMP requires manually specifying predicates and their effects while LLMs generally do not. Yet in this work a significant amount of additional information must be provided to the model, from the descriptions of the skills and examples of their operation to access to a simulator in order to test for constraint satisfaction. It is not entirely clear which is preferable.

**Quality Of The Limitations Section:**

3

**Questions For Rebuttal:**

1. How could a method like this be deployed on robots? Currently it seems that the method requires significant knowledge of the environment or makes assumptions that are generally impractical in the real world including skills that execute perfectly, knowledge of perfect object poses, knowledge of exact constraint violations and access to a physics simulator to actually test if constraints are met or not.
2. Lack of comparison to methods that combine LLMs and task planning, for example https://arxiv.org/pdf/2304.11477
3. No ablations / analysis beyond the main results - Can you probe the method to understand why it works? What is it sensitive to? For example how sensitive is it to the number of samples taken?
4. Appendix lacking descriptions of the skills - how are they implemented, what is their individual success rate in simulation, what sort of information do they require to operate?
5. Suggestion: For table 1, perhaps move your method to the bottom row and also include an average column that shows the average performance improvement
6. What are the failure cases for this method, when it doesn't work is it due to incorrect plans, incorrect continuous parameters or something else?
7. Lack of comparisons to VLM methods which could in principle take into account at least some of the constraints by simply processing the image of the scene.

**Robotics Focus:**

2

**Summary Of Paper:**

This work tackles the problem of constraint satisfaction in the context of LLM planning by integrating in sampling and constraint testing into the LLM planning framework. The results demonstrates its capability on a set of simulated manipulation tasks.

**Summary Of Recommendation:**

Overall, I don't believe this work meets the bar for acceptance at CoRL. While it describes and aims to solve a concrete problem with LLM planning, it is not likely to operate on a robot in this form due to its significant assumptions and there is significant experimentation and analysis missing.

---

### Official Review · Reviewer_tB3j · 2024-07-22
**Review of Trust the PRoC3S**

**Originality:** 3
**Technical Quality:** 3
**Clarity Of Presentation:** 3
**Potential Impact:** 2
**Recommendation:** 3
**Confidence:** 4

**Review:**

**Pros:**

The paper aims to address a challenging problem (long horizon manipulation) in robot manipulation. It assumes a primitive set of skills and focuses on how to chain them such that the task can be accomplished and all the constraints can be satisfied. Overall, the paper is well written and the approach makes sense. Experiments seem to show that the method works much better than baselines.

In doing the latter access to a simulator is assumed which can observe skill effects as well as constraint violation (using constraint violation functions).

**Cons and Questions:**

Baselines: It seems that the proposed approach (with its model based reasoning from the simulator) works much better than model-free LLM approaches (such as CaP). Are these differences because the tasks require complex model based reasoning (i.e. the depth of the search tree to find the optimal plan can be large) or is it just that without any checks for constraint satisfaction, CaP based approaches often output plans that end up violating states.

Optimality:  A nice thing about the TAMP framework, which is clearly a heavy inspiration for the proposed approach is that we can get optimal plans since a discrete planner can optimize the plan cost. But this seems to be much harder to achieve in the current framework. This is because the LLM outputs some plan and then constraint checking happens on this plan, and the output of this will be sent back to the LLM to refine its initial plan. It is unclear to me how long can this feedback loop last. Did the authors perform experiments on this? I would imagine that for some complex geometric tasks the LLM can completely fail beyond a point and it may not have back-tracking etc ability like a true planner does. Basically, with this approach we seem to be inheriting a lot of pitfalls of LLM as planners. Maybe the paper should discuss this in the limitation section. Similarly, the optimality of the final generated plans should be discussed.

Similarly, using a normal TAMP pipeline seems like a very reasonable approach for the experiments considered here. Is there some particular reason the authors did not add that as a baseline? Interestingly, the no-feedback (PRoC3S-NF) baseline performs quite well on most tasks almost same as the proposed approach, failing on some other tasks. For these other tasks, where this approach fails how many rounds of feedback with the LLM is required. Quantitative analysis of this kind would be very useful to understand how complex the provided domains and problems are.

**Quality Of The Limitations Section:**

3

**Questions For Rebuttal:**

see above

**Robotics Focus:**

3

**Summary Of Paper:**

This paper proposes an LLM based planning approach to solve long horizon manipulation tasks. The planning part of the approach is somewhat similar to TAMP approaches, however, with certain differences. Overall, in this proposed approach, an LLM is used to generate a function that accepts some open parameters and outputs grounded skills. Together with this function, the approach outputs another function  from which continuous parameters can be sampled. Once both of these functions are returned by the LLM, a simple sample-and-test procedure is applied to check if the returned list of skills together with sampled parmaeters can reach goal states without violating any constraints. If not, some information (such as description of most common violations etc) are  returned to the LLM which is asked to provide a new plan for the task.  Experiments  are performed on 3 different table-top environment settings.

**Summary Of Recommendation:**

The paper is well written and aims to provide an interesting new approach. However, experimental details are very limited and its hard to say how well does the approach work compared to tranditional tamp approaches.

---

### Official Review · Reviewer_7svD · 2024-07-24
**Well written paper, but assumptions seem a bit strong to render this method useful on actual robots.**

**Originality:** 2
**Technical Quality:** 3
**Clarity Of Presentation:** 3
**Potential Impact:** 2
**Recommendation:** 2
**Confidence:** 4

**Review:**

## Strengths
1. The paper clearly defines the set of assumptions and input/output contract of the model. Overall, the paper is well written and generally free of typos/grammatical errors. The paper has clear figures that assist understanding the point that the authors are making.
2. The baselines chosen in the experiment are clearly described, and cover a good number of the design decisions of the main algorithm.
3. The authors clearly highlight their method's differences from prior work. Specifically, incorporating feedback from constraint violations into future replans is an interesting approach to TAMP.

## Weaknesses
1. At a broad level, the paper seems to make a set of rather restrictive assumptions: full observability, access to constraint functions, and access to a model that *describes* the types of failure modes in the model. It is not immediately obvious how especially the latter would be available when applying this method on real robots.
2. Ln 190-191: I do not follow the claim that the code-generation capabilities of LLMs helps reduce the reasoning to two-parameter space. It seems that this is true only for the running example, but cannot be stated as true in general. For example, inserting an arrow-shaped block into a mold requires both position and rotational deltas.
3. Assuming constraints to be a set of functions that return only binary values seems to be a rather strong assumption. While some constraints may fall under this paradigm, many real-world constraints (especially if dealing with human users) may be more appropriately expressed as a continuous scale for which tradeoffs can be made.
4. The experiment section would benefit from additional metrics/runs. Currently, only success rate is reported as a metric, but there are components of the proposed algorithm (e.g. resampling budget, average number of times to resample, inference speed) that are left untested. Furthermore, experiments are done in only 3 simulated environments. It would be nice to see extensions to a couple more tasks and/or different robots, which would help emphasize the proposed method's ability to respect kinematic constraints.
## Low-level technical:
* Ln 188-189: Authors mention "...pick and place skill requires six continuous parameters", but only 3 are specified.

**Quality Of The Limitations Section:**

3

**Questions For Rebuttal:**

1. Ln 189-191: Can the authors please clarify what they mean by "the generated LMP reduces the sampling space to two-parameter space"? It is not clear whether the parameters are referring to dx, dy. As mentioned in W1 above, can the authors please clarify why this is related to the code generation capabilities of LLMs?
2. While this paper mainly address constraints with respect to collisions, how would this method handle human preferences? For example, if the task description has multiple possible final goal configurations, or *no* possible goal configuration, both of which may require access to human preferences, how would this framework address the issue?
3. Some of the "constraints" illustrated in Figure 1, such as "Grasp Failure" seem to be more attributed to test-time perturbations or error. In other words, while the robot may execute a suboptimal grasp some percentage of times and "violate" a constraint, this does not necessarily warrant a change in the plan. Can the authors please clarify the distinction on this point?
4. Can the authors please clarify the definition of "continuous" parameters and why this is a traditionally hard problem? It seems as if reasoning and satisfying continuous constraints is a main contribution of the paper, but the inherent challenges in predicting/reasoning in such continuous spaces is a bit vague. What are examples of inputs/outputs that are not continuous?
5. Can the authors provide more description as to the simulated world model in which they perform test rollouts? How is it different from the actual simulator in which evaluations is done?
6. What were the prominent failure modes of the proposed algorithm in the experiments that were run?

**Robotics Focus:**

2

**Summary Of Paper:**

This paper proposes to satisfy environment constraints in TAMP by evaluating a task list against a set of constraint functions, and replanning based on language descriptions of how the constraint was violated.

**Summary Of Recommendation:**

The method is sound, and assumptions/limitations are outlined clearly. However, the assumptions are sufficiently restrictive such that it is not obvious to adapt such a method to hardware. Given such, I don't believe the proposed method would be very impactful in the robot learning community.

---

### Author Rebuttal · Authors · 2024-08-14

In our attached rebuttal file, we include an updated draft of our paper with changes highlighted in red, our original supplementary video, as well as new supplementary video showcasing our new real-robot tasks.

---

### Decision · Program_Chairs · 2024-09-04

**Decision:**

Accept

**Comment:**

Update: The authors provided sufficiently detailed responses in the rebuttals that the reviewers found satisfactory. The idea of using a simulator to act as a "testing ground" for LLMs to revise its outputs before deploying on a real robot is worth sharing more broadly with the CoRL community. I am recommending to accept this submission.

Original: The reviewers agree that the paper is well written and tackles an important problem. However, they are concerned about the strict assumptions the proposed method makes as well as a lack of convincing baselines. I ask the authors to carefully address all reviewers' feedback during the rebuttal period.